# Establishment of a Suspension MDBK Cell Line in Serum-Free Medium for Production of Bovine Alphaherpesvirus-1

**DOI:** 10.3390/vaccines9091006

**Published:** 2021-09-09

**Authors:** Pengpeng Wang, Shulin Huang, Chengwu Hao, Zhanhui Wang, Haoran Zhao, Mengyao Liu, Xinrui Tian, Letu Ge, Wenxue Wu, Chen Peng

**Affiliations:** 1Key Laboratory of Animal Epidemiology and Zoonosis, College of Veterinary Medicine, China Agricultural University, Beijing 100193, China; 18754881870@163.com (P.W.); wangzhanhui2016@163.com (Z.W.); zhaohrr@yeah.net (H.Z.); mengyaoliu@cau.edu.cn (M.L.); tianxrui@163.com (X.T.); 17600366745@163.com (L.G.); 2China Animal Husbandry Industry Co., Ltd., Beijing 100070, China; hslcau@163.com; 3TECON Biopharmaceutical Co., Ltd., Urumqi 830011, China; haochengwu@tecon-bio.com

**Keywords:** serum-free suspension adaptation process, suspended MDBK cells, serum-free medium, bovine alphaherpesvirus-1 (BoHV-1), virus production, ultrasonic treatment

## Abstract

The Madin–Darby bovine kidney (MDBK) cell line is currently used for the production of bovine alphaherpesvirus-1 (BoHV-1) vaccine. For the purpose of vaccine manufacturing, suspension cells are preferred over adherent ones due to simplified sub-cultivation and an easier scale-up process, both of which could significantly reduce production cost. This study aimed to establish a procedure for the culture of BoHV-1 in the suspended MDBK cell line in serum-free medium. We screened several commercially available serum-free media and chose ST503 for subsequent experiments. We successfully adapted the adherent MDBK cells to suspended growth in ST503 in the absence of serum. The maximum density of suspension-adapted MDBK cells could reach 2.5 × 10^7^ cells/mL in ST503 medium with optimal conditions. The average size of suspension-adapted cells increased to 18 ± 1 µm from 16 ± 1 µm. Moreover, we examined tumorigenicity of the suspended cells and found no sign of tumorigenicity post adaptation. Next, we developed a protocol for the culture of BoHV-1 in the cell line described above and found that ultrasonic treatment could facilitate virus release and enhance virus yield by 11-fold, with the virus titer reaching 8.0 ± 0.2 log_10_TCID_50_/mL. Most importantly, the prototype inactivated BoHV-1 vaccine we generated using the suspension cultures of MDBK cells induced neutralizing antibodies to a titer comparable to that of the commercial inactivated BoHV-1 vaccine. Overall, we established and optimized a protocol for the production of inactivated BoHV-1 vaccine in MDBK cells adapted for suspension culture, which provides insights for future large-scale manufacturing of BoHV-1 vaccine.

## 1. Introduction

Adaptation of adherent cells to serum-free suspension culture is a procedure widely used in the production of biological products such as vaccines, monoclonal antibodies and pharmaceutical protein products [1,2,3]. Compared to adherent tissue culture, serum-free culture can reduce the cost and improve the quality and stability of biological products. It is well known that animal serum may contain nonspecific inhibitory factors or other biological components that may negatively affect virus replication [4,5], and the residual serum protein in vaccine produced might cause allergic reactions in vaccinated animals [6]. Therefore, media formulated without serum components are widely used for the growth of many cell lines designated for vaccine production [7,8].

The Madin–Darby bovine kidney (MDBK) cell line is an immortal cell line derived from apparently normal bovine kidney and is adherent by nature with medium containing fetal bovine serum [9]. MDBK cells are permissive to a variety of viruses and are usually adherently cultured in roller bottles or on microcarriers for mass production of vaccines in industry [10,11]. It is well accepted that adherent tissue culture requires more steps and is more time and labor intensive than suspension culture. Compared to adherent tissue culture, suspension culture is more space efficient and can support the growth of cells with higher concentration, which is more suitable for industrial production. Meanwhile, studies have shown that production of bovine adenoviruses (BAdV) in suspended MDBK cells led to higher viral yields than that in adherent cells [12].

Bovine respiratory disease complex (BRDC) is a general term for cattle respiratory diseases caused by a variety of viruses, bacteria and other factors [13]. Because BRDC is a multifactorial disease, it is often persistent and difficult to resolve due to the unique immunosuppressive and immune-avoidant mechanisms exhibited by each pathogen, and co-infection of multiple pathogens can enhance the toxicity and persistence of virus and bacteria [14]. Bovine alphaherpesvirus-1 (BoHV-1) is one of the viruses that is strongly associated with bovine respiratory disease complex (BRDC), which affects cattle health worldwide and causes enormous economic losses in the cattle industry [15]. The virus has been widely spread and reported cases have increased rapidly since the virus was first isolated in 1956 [16]. Studies have shown that the positive rates of BoHV-1 antibody in cattle herds in Ireland, China and Brazil could reach 75–90%, 37.9% and 84%, respectively, and milk yield of the cattle infected with BoHV-1 will be reduced by nearly 250.9 L per year [17,18,19,20]. The virus can cause a variety of clinical symptoms in cattle, including infectious bovine rhinotracheitis (IBR), conjunctivitis and abortion. Vaccination remains one of the most effective strategies to prevent and control the disease. Currently, the vaccine of BoHV-1 is manufactured in adherent MDBK cells with limited production efficiency due to the limitations of adherent tissue culture. Therefore, the adaptation of MDBK cells to suspension culture and the production of BoHV-1 were studied. We screened a number of commercially available serum-free media and identified one suitable for the culture of MDBK cells. Next, we adapted adherent MDBK cells to suspended growth in serum-free medium, and evaluated the proliferation and tumorigenicity of suspended MDBK cells. Finally, we established a protocol for the production of BoHV-1 in suspension MDBK cells cultured in shake flask. The production of BoHV-1 in suspension cultures can be improved with the enhancement of cell density, which has provided obvious advantages over adherent cell culture. Importantly, the BoHV-1 produced in suspension culture exhibited satisfactory quality and immunogenicity.

## 2. Materials and Methods

### 2.1. Cell Lines, Culture Media and Animals

The adherent MDBK cells (NBL-1, #CL21), adherent Hela cells (#CL16) and adherent ST cells (#CL27) were obtained from the China Veterinary Culture Collection Center (CVCC, Beijing, China). These adherent cells were cultivated in Dulbecco’s modified Eagle medium (DMEM) (Gibco Invitrogen, USA) supplemented with 10% (*v/v*) fetal bovine serum (FBS) (BI, Israel) and 1% (*v/v*) penicillin–streptomycin liquid (PS) (Solarbio, Beijing, China). Cells were maintained in a humidified incubator with 5% CO_2_ at 37 °C.

All animal experiments were performed in accordance with the China Agricultural University Institutional Animal Care and Use Committee guidelines (AW30601202-2-1) and with the International Guiding Principles for Biomedical Research Involving Animals. Experiments were approved by the Beijing Administration Committee of Laboratory Animals. Twelve female nude (nu/nu) athymic mice aged 4 to 5 weeks were obtained from Charles River Laboratories (China), and fifteen female guinea pigs (aged 5 to 6 weeks) used in this study were obtained from Xinglong experimental animal farm, Haidian District, Beijing (China).

### 2.2. Cell Counting

Adherent cells were washed three times with PBS and then treated in 1 mL of 0.25% trypsin–EDTA (Gibico, Waltham, MA, USA) followed by 3 min incubations at 37 °C. For suspension cells, the cell suspension was directly used for counting.

Trypsinized or suspended cells were mixed with trypan blue solution (0.4%) (Solarbio, Beijing, China) in equal volume and stained for 3 to 5 min prior to counting. Cell counting and viability assessment were performed with an automated cell counter (Thermo Fisher Scientific, Waltham, MA, USA).

### 2.3. Selection of Suspension Medium

Five commercial serum-free media were tested for the adaption process. These media were ST503 (Vbiosci, Beijing, China), CD MDBK 249 (JS Biosciences, Shanghai, China), OPM-MDBK SFM1 (OPM Biosciences, Shanghai, China), Uni101 (Vbiosci, Beijing, China) and Bofit-S002 (Bioengine, Shanghai, China). After mixing with DMEM in equal volume and being supplemented with 10% (*v/v*) FBS and 1% (*v/v*) PS, they were designated A–E in the paper, respectively. Adherent MDBK cells at the initial density of 1.0 × 10^6^ cells/mL were incubated with the above media. These cells were cultured in a 125 mL shake flask (Corning, New York, NY, USA). The culture volume was 20 to 30 mL and agitation was 100 rpm at 37 °C with 5% CO_2_. The number and viability of cells were assessed and medium replaced every 2 or 3 days. Medium with the highest rate of cell viability was selected for the subsequent suspension adaptation process of MDBK cells.

### 2.4. Adaptation of Adherent MDBK Cells to Grow in Suspension and Serum-Free Culture

Adherent MDBK cells were cultivated in suspension with suitable serum-free medium in a 125 mL shake flask at a concentration of 1.0 × 10^6^ cells/mL to start the suspension adaptation process. The medium was supplemented with 3% (*v/v*) FBS and 1% (*v/v*) PS and cells were kept in a humidified incubator with 5% CO_2_ and at 37 °C. The culture volume was 20 to 30 mL and agitation was 100 rpm. The number and viability of cells were assessed every 24 h and cells were passaged every 2 or 3 days. Once the cells were adapted to grow in medium with 3% serum in suspension, medium was discarded and replaced with serum-free medium for suspension adaptation. All passages of suspension-adapted cells were frozen with medium containing 10% (*v/v*) FBS as well as 10% (*v/v*) dimethyl sulfoxide (DMSO) (Solarbio, Beijing, China) and stored in liquid nitrogen for future evaluation.

### 2.5. Establishment of Growth Curve of Suspension-Adapted MDBK Cells

MDBK cells grown in suspension were inoculated in a 125 mL shake flask at a concentration of 1.0 × 10^6^ cells/mL and incubated with agitation in an incubator with 5% CO_2_ at 37 °C. Cell viability and number were evaluated every 24 h to establish the growth curve from 0 to 144 h. All experiments were performed in triplicate.

### 2.6. Examination of Tumorigenicity for Suspension-Adapted MDBK Cells

The protocol for examining tumorigenicity was formulated based on the “Veterinary Pharmacopoeia of the People’s Republic of China, CVP”. Suspended MDBK cells were quantitated and used to inject into nude (nu/nu) athymic mice to evaluate the tumorigenicity. There were four experimental groups in the tumorigenicity experiment, with three athymic nude (nu/nu) mice (4 to 6 months old) in each group. Mice in Group 1 were injected subcutaneously in the subaxillary area with 1.0 × 10^7^ suspended MDBK cells diluted in 0.1 mL of serum-free medium. Mice in Group 2 and Group 3 were injected with 1.0 × 10^6^ Hela cells or 1.0 × 10^7^ ST cells, respectively. Mice in Group 4 were injected with 0.1 mL of serum-free medium only. All mice were observed daily and the injection sites were palpated for lesion development for up to 16 weeks. Tissue samples from the gross lesions and injection sites of all mice were collected for histological examination. Specifically, collected tissues were maintained in 4% paraformaldehyde solution, embedded in paraffin, cut into sections with the thickness of 6 μm and then placed onto positively charged glass slides prior to staining with hematoxylin and eosin (H&E) according to a protocol described previously [21].

### 2.7. Virus Strain and Virus Titration

The SZH strain of BoHV-1 preserved in our laboratory was used in this study. The strain was isolated from cattle in Beijing and had good immunogenicity, and could be used as a candidate strain for a BoHV-1 vaccine. Viral titers were determined by 50% tissue culture infective dose (TCID_50_) on adherent MDBK cells and were calculated by the Reed–Muench method, represented as TCID_50_/mL [22].

### 2.8. Optimization of BoHV-1 Production in Suspension-Adapted MDBK Cells

BoHV-1 was passaged 3 times in suspended MDBK cells, quantitated and designated as the seed virus. Suspension-adapted MDBK cells at the densities of 2.0 × 10^6^ cells/mL, 4.0 × 10^6^ cells/mL and 8.0 × 10^6^ cells/mL were infected with the seed virus at the multiplicity of infection (MOI) of 0.1, 0.05 or 0.01. Cells were then harvested with medium at different time points post infection, frozen and thawed 3 times, and the supernatant was stored for virus titration with the protocol described above.

Furthermore, the following methods for virus release were compared for their effects on the recovery of viruses from the MDBK cells. (1) Repeated freeze–thaw cycles (freezing at −80 °C, thawing at room temperature). (2) Ultrasonic treatment. The sonication was performed with the ultrasonic cell grinder (Scientz, Ningbo, China) and the output power was 180 W. The working volume was 0.5 to 1.0 mL, and the infected culture was treated for 1 to 10 min. Additionally, the treated virus suspension was then centrifuged at 3000 rpm for 5 min and the supernatant was filtered by 0.2 μm filter (Pall Corporation, Port Washington, NY, USA) for virus titer determination.

### 2.9. Production of BoHV-1 in Adherent MDBK Cells

Adherent MDBK cells were grown in a 150 mm cell culture dish (Corning, New York, NY, USA). The cells in the last phase of logarithmic growth were washed three times with PBS and 20 mL of DMEM (containing 1% PS) was added. The virus was added at MOI of 0.1. After absorption for 1 h at 37 °C, the cells were washed with PBS, and 30 mL of DMEM (containing 2% FBS and 1% PS) was added. After incubation for 36 h at 37 °C, the infected culture was subjected to ultrasonic treatment. As described above, the treated virus suspension was centrifuged and filtered for virus titration.

### 2.10. Immunogenicity of BoHV-1

Viruses were inactivated by the treatment of diethyleneimine (BEI) (Solarbio, Beijing, China) with the final concentration of 2 mmol/L for 24 h at 37 °C with agitation at 100 rpm. Next, the sodium thiosulfate pentahydrate (Solarbio, Beijing, China) at the final concentration of 2% was added to terminate the inactivation process. The inactivated virus suspension was mixed 1:1 (*v/v*) with Montanide ISA 201 VG (Seppic, Paris, France), vortexed for 5 min and then kept at room temperature for 1 h. Fifteen healthy guinea pigs were randomly divided into three groups (5 per group). One group was immunized with ISA 201 adjuvant alone, and the other two groups were immunized with inactivated BoHV-1 produced in suspended MDBK cells or purchased from Jinyu Group (Lot number 180215019549) with the same recipe for adjuvant. Antigens were quantified according to the virus titer. Each guinea pig was immunized with the same amount of antigen by intramuscular injection on day 0 and boosted on day 21 with the same antigen. Fourteen days after the boost, blood samples were collected and titers of the neutralization antibodies were determined by the protocol described previously by Quattrocchi et al. [23].

### 2.11. Statistical Analysis

Results are represented as mean ± the standard deviation (SD). GraphPad Prism program was used for data visualization. Student’s *t*-test was performed to assess statistical significance (cell sizes, neutralization antibody titers and virus titers).

## 3. Results

### 3.1. Screening of Commercially Available Medium for Suspended Growth of MDBK Cells

We first tested five commercially available media supplemented with 10% (*v/v*) FBS for their capacity to support MDBK cell growth in suspension culture. MDBK cells maintained in DMEM medium were trypsinized, resuspended and grown in a 125 mL shake flask using the following media: ST503 (A), CD MDBK 249 (B), OPM-MDBK SFM1 (C), Uni101 (D) and Bofit-S002 (E), and were mixed with DMEM in equal volume. Cells were passaged every 2 days and monitored daily for cell viability and morphological changes. While cells grown in A, B and C appeared to be mostly dispersed, cells grown in D and E tended to aggregate over time, and media D and E were thus deemed unsuitable for suspension culture (Figure 1A–D). Cell proliferation was quantified in media A, B and C for up to three passages and growth curves were graphed and presented in Figure 1F. Cell growth halted in medium B after passage 2 and live cell numbers declined rapidly from the beginning of the culture for medium C. In comparison, the growth of cells fluctuated for the first two passages in medium A, a sign of adaptation in new medium, and then began to proliferate steadily after passage 3 (not shown). Cells grown in medium A also displayed the highest level of viability among all five groups tested and ST503 was thus chosen for the subsequent adaptation.

### 3.2. Adaptation of MDBK Cells to Suspension Culture in Serum-Free Medium

To prepare suspended MDBK cells for growth in serum-free medium, we gradually reduced serum concentration in the medium and at the same time began to grow cells in suspension. We first lowered the serum content (FBS) from 10% to 3% (*v/v*) and grew cells in 125 mL shake flasks with agitation (100 rpm). Cells were passaged every 2 days for 17 passages and a portion of cells from each passage was frozen for future experiments. As shown in Figure 2A, viable cell concentration nearly doubled after only one passage and remained between 2.0 × 10^6^ and 3.0 × 10^6^ for the subsequent passages. On the other hand, cell viability plummeted in the first three passages but began to recover after passage 4 until it became relatively steady after passage 6 (Figure 2B). In addition, no cell aggregates were found after passage 6. Next, cells from the 13th passage were thawed and revived to grow in the absence of serum in ST503 for serum-free adaptation. Cells were maintained in serum-free ST501 with agitation at 37 °C, 5% CO_2_ and passaged every 2 days. Viable cell concentrations and viability were monitored regularly and are shown in Figure 2C,D, respectively. Viable cell concentration increased to a level above 2.0 × 10^6^ and remained steady throughout the adaptation process, while oscillation of cell viability was observed until after passage 25. Overall, both cell viability and concentration were steady after serum-free adaptation and no cell aggregation was seen for the adapted cells.

### 3.3. Characterization of Suspension-Adapted MDBK Cells

In order to identify biological characteristics of suspension-adapted MDBK cells, we examined the morphological changes, tumorigenicity and diameter of suspension-adapted MDBK cells and established the growth curve of suspension-adapted MDBK cells. Compared to adherent MDBK cells (Figure 3A), suspension-adapted cells appeared rounded and buoyant (Figure 3B). Diameters of both cell types were measured with an automated cell counter and the suspension-adapted cells (18.3 ± 1.3 µm) were enlarged through the adaptation process compared to the adherent ones (16.5 ± 0.9 µm) (*p* < 0.05, Figure 3D).

To establish the growth curve of suspension-adapted MDBK cells, we determined cell viability and quantity at various time points. The suspended MDBK cell seeds frozen in liquid nitrogen were recovered, and the cell viability was maintained above 92%. After three passages of adaptation, the suspension MDBK cells were inoculated into the ST503 medium with an initial density of 1.0 × 10^6^ cells/mL. Cell viability and number were evaluated every 24 h. As shown in Figure 3C, cell growth entered the log phase at 24 h and reached the plateau at 72 h when the maximal cell density was at 8.5 × 10^6^ cells/mL. After 120 h of cultivation, both cell viability and number began to decrease. In order to elucidate if the cessation of cell growth was due to exhausted nutrients in the medium, ST503 medium was replaced at 72 h to make up for the effects of nutrient deficiency and metabolite accumulation. As shown in Figure 3C, cell proliferation continued after medium change and remained at a high growth rate. The maximum cell density at 144 h reached 2.5 × 10^7^ cells/mL, which showed that the suspended MDBK cells displayed a robust proliferative potential.

To evaluate if suspension and serum-free adaptation triggered cell tumorigenicity, we elucidated if the adapted cells could cause tumors in mice. Nude mice were injected subcutaneously with 1.4 × 10^7^ suspension-adapted MDBK cells (Group 1), 1.5 × 10^6^ Hela cells (Group 2), 1.2 × 10^7^ ST cells (Group 3) or medium only (Group 4). When tumor progression was monitored at 14 days post injection, solid tumors had formed in the Hela-cell-injected group, the positive control group, and the pathological section of the skin at the injection sites showed that there were dense tumor cells arranged subcutaneously with clear boundaries with the surrounding tissues, and a large number of inflammatory cells were enriched around the tumor tissue (Figure 4B). None of the group 1, group 3 and group 4 mice showed any sign of tumor formation at the injection site. These mice were sacrificed for pathological and histological examination at 16 weeks post injection. The results of pathological necropsy showed that no nodules were found in any lymph nodes and organs, and it was observed by microscope that there were no tumor cells and only a small amount of inflammatory cell infiltration at the injection site in group 1, group 3 and group 4 (Figure 4A,C,D). In conclusion, the suspension-adapted MDBK cells showed no sign of tumorigenicity.

### 3.4. Production of BoHV-1 in Suspension-Adapted MDBK Cells

To establish a protocol for producing BoHV-1 in the suspension-adapted MDBK cells, we first serially passaged the BoHV-1 SZH strain in suspended MDBK cells three times. Suspended MDBK cells at 8.0 × 10^6^ cells/mL were infected with BoHV-1 SZH at the MOI of 0.1 and viruses were harvested at several time points and quantitated. Replication curves of viruses from three passages were graphed and shown in Figure 5A. Viruses from passage 3 replicated to a titer of 7.3 log_10_TCID_50_/mL, which was about 4-fold higher than that of passage 1 and was used for further optimization of the protocol. We next evaluated the effect of starting cell concentration on the replication of BoHV-1 in suspended MDBK cells by infecting cells at different densities with BoHV-1 at three different MOIs.

The suspended MDBK cells were cultured in 125 mL shake flasks at the initial densities of 2.0 × 10^6^ cells/mL, 4.0 × 10^6^ cells/mL or 8.0 × 10^6^ cells/mL, and were infected with BoHV-1 at the three different MOIs (0.01, 0.05 or 0.1). Cells were harvested at different time points post infection, frozen and thawed repeatedly, and virus was titrated. For the cell concentrations 2.0 × 10^6^ and 4.0 × 10^6^ cells/mL (Figure 5B,C), higher MOI infections could result in higher virus yield. In comparison, when the cell concentration was 8.0 × 10^6^ cells/mL (Figure 5D), the highest yield (6.4 ± 0.4 log_10_TCID_50_/mL) obtained was from cells infected at MOI of 0.05. From these optimal results (Figure 5E), we found that the highest viral yield was obtained with virus infection of 0.05 MOI at the cell density of 8.0 × 10^6^ cells/mL. Although the differences were not significant, the higher size of cell inoculums was more likely to obtain a higher virus yield. Additionally, the increasing trend of virus yield was more obvious with the increase in cell density when the MOI was higher (Figure 5E). We also examined if the method of virus release directly affected virus yield. Total viruses produced from suspended MDBK cells were either released by repeated freeze–thaw cycles (FT) or ultrasonic treatment (UT) and the virus yields were determined by TCID_50_ on adherent MDBK cells and graphed (Figure 5F). For the UT groups, the highest viral yield was observed when cells were sonicated for 1 min, and prolonged treatment did not further increase the virus yield. The virus yield in the FT group was significantly lower than that of the UT group.

Next, we compared virus yields from suspended cells with that from adherent cells. Both adherent and suspension cultures were infected at a density of 0.9 × 10^6^ cells/mL with an MOI of 0.1. The data in Figure 5G show that the virus productivity of the adherent MDBK cell (7.6 ± 0.1 log_10_TCID_50_/mL) was higher than that achieved by the suspension MDBK cell (6.0 ± 0.2 log_10_TCID_50_/mL) in a shake flask. However, the virus productivity increased to 8.0 ± 0.2 log_10_TCID_50_/mL in the suspension culture when the cell density at infection increased to 8.0 × 10^6^ cells/mL from 0.9 × 10^6^ cells/mL. These results indicate that although the virus productivity of suspension MDBK cells was lower than that of adherent MDBK cells, we could increase the production of BoHV-1 by increasing the initial cell density, which is only possible with suspended MDBK cells.

### 3.5. Immunogenicity of Inactivated BoHV-1 Produced in Suspended MDBK Cells

To assess if cultivating BoHV-1 in suspended MDBK cells altered the immunogenicity of inactivated BoHV-1 vaccine, we compared inactivated BoHV-1 made in suspended MDBK cells to the commercial inactivated BoHV-1 vaccine we purchased from Jinyu Group. Guinea pigs weighing 350–400 g were divided into three groups and animals were immunized with the adjuvant only, inactivated BoHV-1 made in suspended MDBK cells or the commercially available BoHV-1 vaccine. The antigen amount of both vaccines was 6.7 log_10_TCID_50_/mL. All animals were boosted with the same antigen at 14 days post primary immunization before serum samples were collected for neutralization antibody assessment. As shown in Figure 6, the adjuvant alone did not trigger antibody production and the commercial vaccine induced high levels of neutralization antibodies specific for BoHV-1. Importantly, the BoHV-1 inactivated vaccine manufactured in suspended MDBK cells could induce neutralizing antibodies at levels similar to those of commercial inactivated BoHV-1 vaccine, and there was no significant difference between the two groups, which suggests that the BoHV-1 in suspension cultures has good immunogenicity.

## 4. Discussion

The suspension adaptation process of adherent cells is mainly to resist anoikis through manual intervention [24]. In general, the process of suspension adaptation can be divided into three steps: serum-free adaptation culture, suspension adaptation culture and bioreactor high-density adaptation culture [25], which is easy to operate and is often applied in the suspension adaptation process of various mammalian cells. For now, a variety of cells, including MDCK cells [26], CHO cells [27], HEK 293SF cells [2], BHK21 cells [28] and Vero cells [29], have been successfully adapted to suspended growth in serum-free medium, and have been used in the industrial mass production of many biological products.

In this study, we adapted adherent MDBK cells to suspension culture in the absence of serum content in two steps. First, cells were adapted to proliferate in medium containing low content of serum in suspension. Secondly, cells were adapted to grow without serum in suspension (Figure 2). Compared with the method established by M.S. Sinacore et al. [25], the two-step process in our study can save the time of suspension adaptation, and simplify the process of suspension adaptation, which can retain the production capacity of MDBK cells and save economic costs. Investigators previously used CD medium from Gibco and CD MDBK 249 medium from JS Biosciences for suspension adaptation of MDBK cells, and were able to obtain densities of only 4.0 × 10^6^ cells/mL and 1.25 × 10^7^ cells/mL, respectively [30,31]. However, we found ST503 medium to be more suitable for suspension growth of MDBK cells, enabling them to reach 2.5 × 10^7^ cells/mL after replacement of depleted medium (Figure 3C). These results indicate that the suspended MDBK cells display a robust proliferative potential.

In the process of suspension adaptation, continuous cell lines may obtain different biological characteristics from the original cells, such as morphology, size and tumorigenicity. As the data show, the morphology and diameter of MDBK cells were changed after the suspension adaptation process (Figure 3A,B,D). Suspended MDBK cells appeared rounded and floated, which could reduce the contact and friction between cells, thus maintaining the integrity and activity of cells. In addition, the limited growth space and the phenomenon of contact inhibition in adherent cultures will affect the growth and size of cells [32], but there are no such restrictions in suspension cultures, so that the average size of suspension cells is larger than that of adherent cells (Figure 3D). Additionally, the continuous cell lines with tumorigenicity as production cells will have a negative impact on the safety of their biological products. Studies have shown that adherent MDBK cells [33] and ST cells [34] are not tumorigenic, while Hela cells can form tumors after being inoculated into nude mice [35]. Therefore, we set Hela cells as the positive control, ST cells as the negative control and ST503 medium as the control group to assess the tumorigenicity of suspended MDBK cells. Suspension-adapted MDBK cells were not found to be tumorigenic from the test results (Figure 4), which demonstrates that the application of suspended MDBK cells in BoHV-1 vaccine can eliminate some hidden dangers and ensure the safety of vaccine. In addition, following authentication of the suspended MDBK cell line and its parental cell line and analysis of their genetic stability in future work, the suspended MDBK cell line will be deposited in CVCC and/or other repositories.

In view of establishing a suitable protocol of BoHV-1 production in suspension cultures, we evaluated the effects of cell inoculation density and MOI on virus yield. We found that there was no significant difference in virus yield among different MOIs (significance analysis not shown), and the lower the cell inoculation density, the weaker the effect of MOI on the virus yield (Figure 5B–D). In general, the virus yield is more closely related to the cell concentration. Some studies have shown that an increased cell density at infection time results in higher viral titers [36], but some scholars have found that increasing cell concentration cannot significantly improve the virus titer, but on the contrary, will increase yield variability [37]. Therefore, according to the growth curve of suspended MDBK cells, the cell densities at the starting point (2.0 × 10^6^ cells/mL), midpoint (4.0 × 10^6^ cells/mL) and terminal point (8.0 × 10^6^ cells/mL) of the logarithmic growth phase were used as experimental conditions to study the effect of cell concentration on virus yield (Figure 3C). Our results show that there was a positive correlation between cell concentration and virus production (Figure 5E), but the increasing trend of virus yield was not obvious. We hypothesized that the reason was that the virus could not be sufficiently released by freeze–thaw treatment. Therefore, we evaluated the effect of different virus release methods on virus recovery. As shown in Figure 5F, the virus yield in the UT group was significantly higher than that of the FT group, and the highest virus titer in the UT group was nearly 11-fold higher than that in the FT group. We also found that prolonging the time of sonication had little effect on the yield of virus, but increasing the number of freezing and thawing cycles would decrease the yield of virus, which might be due to the more negative effect of freeze–thaw treatment for the activity of extracellular virus than that of ultrasonic treatment. After ultrasonic treatment, the virus yield could be enhanced by nearly 100-fold when the cell concentration increased from 0.9 × 10^6^ cells/mL to 8.0 × 10^6^ cells/mL (Figure 5G), which indicated that we could obtain higher virus yield by increasing cell inoculation density.

We also compared the virus productivity of adherent and suspension cells, and found that the production capacity of suspended MDBK cells was insufficient (Figure 5G). This may be due to a change in expression of some cell proteins in the suspension adaptation process, which indirectly affects the replication of virus. Some scholars found that compared with adherent cells, the expression of vimentin in suspension cells decreased [38]. Because vimentin is involved in the infection and replication of virus in cells [39], its decrease in expression affects the replication of virus. This might explain the defect of suspended MDBK cells in replication of virus. However, the virus yield could be improved and increase in suspended MDBK cell concentration, and we could obtain a higher concentration of suspended MDBK cells and a larger volume of culture than adherent culture, which could make up for the defect of suspension MDBK cells to a certain extent. In addition, when the immunogenicity of BoHV-1 cultivated in suspended MDBK cells was evaluated, the level of neutralizing antibody induced by suspension BoHV-1 inactivated vaccine was similar to that of the commercial inactivated BoHV-1 vaccine (Figure 6). In a previous study, we compared the level of neutralizing antibodies induced by the commercial BoHV-1 inactivated vaccine and inactivated BoHV-1 in adherent culture, and found there was no significant difference in neutralizing antibody titer between the two groups [40]. Although there were differences among different batches of guinea pigs, it still shows that the BoHV-1 in suspension culture could induce a high level of neutralizing antibody in guinea pigs.

Bovine neonatal pancytopenia (BNP) is an alloimmune syndrome caused by vaccine-induced alloreactive antibodies [41]. Some studies have shown that the MDBK cell line is presumably the origin of BNP-associated alloreactivity [41,42]. However, not all vaccines produced with MDBK cells can cause BNP, and the vaccine leading to BNP may be due to the improvement in the immunogenicity of MDBK cells by adjuvants with efficient and stable conformation [43,44]. Therefore, the antigen purification and adjuvant screening are technical processes that need to be carefully optimized. We will also take whether BNP can be caused by the vaccine as an important index to measure its safety in follow-up research.

## 5. Conclusions

In conclusion, our results demonstrate that adherent MDBK cells were successfully adapted to grow in suspension in ST503 medium. The maximum density of suspension-adapted MDBK cells could reach 2.5 × 10^7^ cells/mL, and they were not found to be tumorigenic. The production protocol of BoHV-1 established in suspension cultures could give full play to the production performance of suspension MDBK cells, and the virus yield could reach 8.0 ± 0.2 log_10_TCID_50_/mL. Furthermore, the BoHV-1 produced in suspension cultures possessed satisfactory immunogenicity. The suspension culture system we established can provide insights for the bioreactor culture of BoHV-1 vaccine, and is expected to be the first step for large-scale culture of other viral vaccines.

## Figures and Tables

**Figure 1 vaccines-09-01006-f001:**
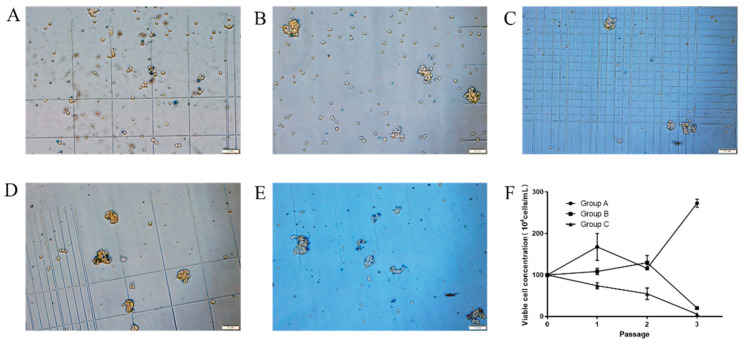
Screening of commercially available media for the suspension adaptation process of MDBK cells. Morphology of MDBK cells in passage 2 maintained in ST503 (**A**), CD MDBK 249 (**B**), OPM-MDBK SFM1 (**C**), Uni101 (**D**) and Bofit-S002 (**E**). Additionally, the growth curve (**F**) of MDBK cells in ST503 (Group A), CD MDBK 249 (Group B) and OPM-MDBK SFM1 (Group C). The scale bar (100 μm) is shown in the lower right corner of the picture (**A**–**E**).

**Figure 2 vaccines-09-01006-f002:**
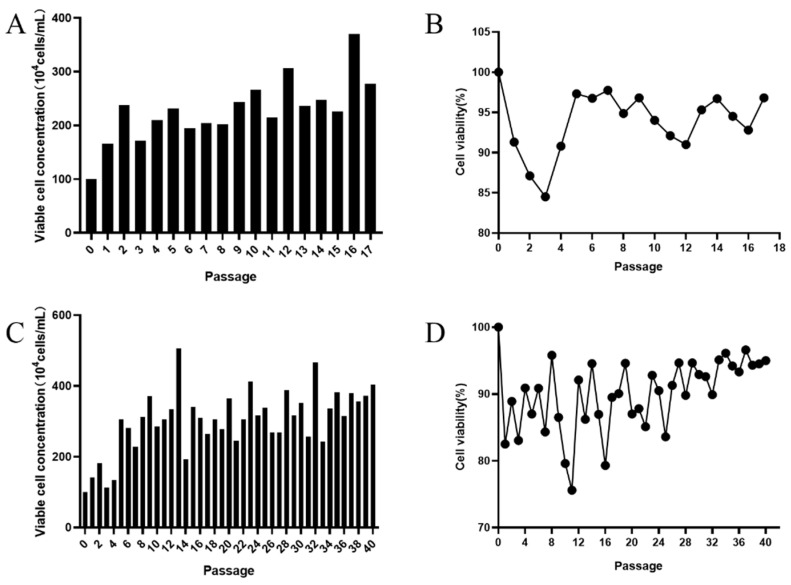
Adaptation of MDBK cells to suspension culture in serum-free medium. The adaptive growth (**A**) and viability (**B**) of adherent MDBK cells in shake flasks with ST503 containing 3% FBS. The adaptive growth (**C**) and viability (**D**) of suspension-adapted MDBK cells in shake flasks with ST503 only.

**Figure 3 vaccines-09-01006-f003:**
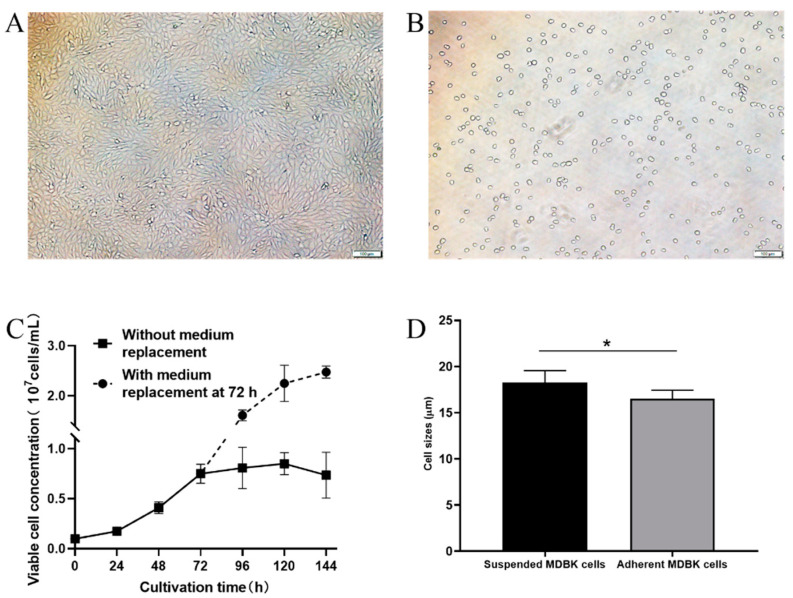
Morphologies of MDBK cells grown as adherent culture (**A**) in T flasks or as suspension culture (**B**) in 125 mL shake flasks. (**C**) The growth curve of suspension MDBK cells. (**D**) Comparison of average size of adherent and suspended cells, significance code: “*” *p* < 0.05. The scale bar (100 μm) is shown in the lower right corner of the picture (**A**,**B**).

**Figure 4 vaccines-09-01006-f004:**
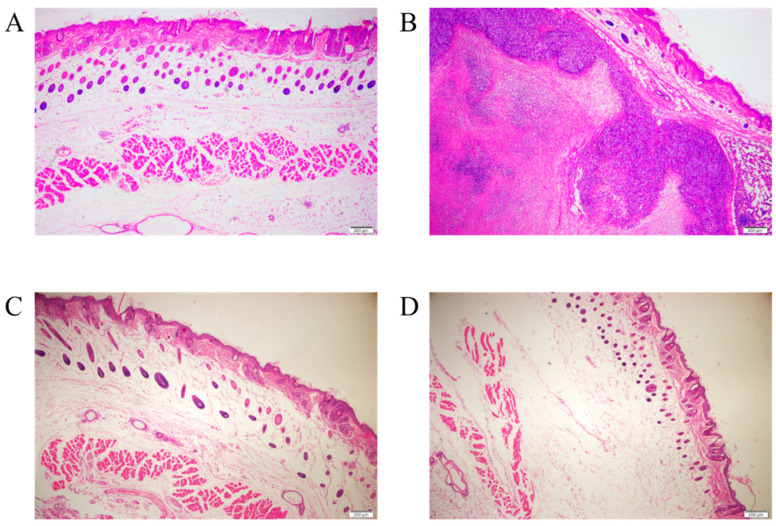
Hematoxylin and eosin stains of the skin tissue from the injection sites of athymus mice. The histological sections (4×) of axillary skin in group 1 injected with suspension MDBK cells (**A**), group 2 injected with Hela cells (**B**), group 3 injected with ST cells (**C**) and group 4 injected with ST503 medium (**D**) were observed by microscope. The scale bar (200 μm) is shown in the lower right corner of the picture.

**Figure 5 vaccines-09-01006-f005:**
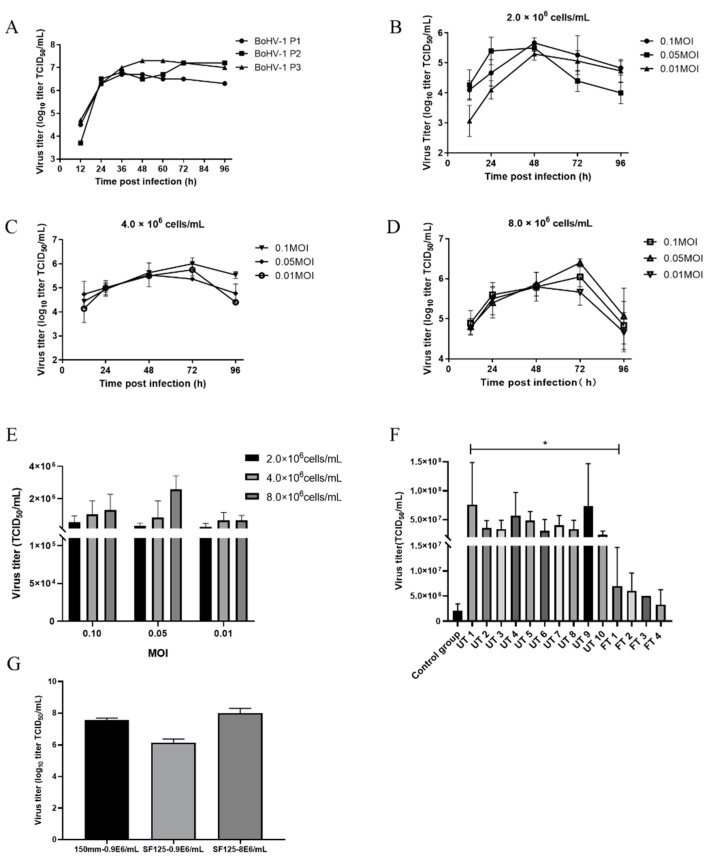
Optimization of BoHV-1 production in suspended MDBK cells cultivated in ST503 medium in shake flasks. (**A**) Virus adaptation: the changes in BoHV-1 titer (in log_10_TCID_50_/mL) after three passages of suspension adaptive culture; the changes in BoHV-1 titer (in log_10_TCID_50_/mL) for the 96 h post infection of suspended MDBK cells at concentrations of (**B**) 2.0 × 10^6^ cells/mL, (**C**) 4.0 × 10^6^ cells/mL and (**D**) 8.0 × 10^6^ cells/mL with BoHV-1 seed virus in shake flasks at MOI of 0.1, 0.05 or 0.01. (**E**) Comparison of the highest virus yield obtained with different cell inoculation concentrations and MOIs. (**F**) Evaluation of different methods of virus release: the titer of infected cultures which were harvested by ultrasonic treatment (UT) for 1 to 10 min (UT 1 to 10), repeated freezing and thawing treatment (FT) 1 to 4 times (FT 1 to 4) or without any virus release method treatment (control group), significance code: “*” *p* < 0.05. (**G**) Production of BoHV-1 in adherent (150 mm cell culture dish) or suspension-adapted (SF125) MDBK cell cultures: the adherent MDBK cells infected at a cell density of 0.9 × 10^6^ cells/mL, and suspension-adapted MDBK cells infected at a cell density of 0.9 × 10^6^ cells/mL or 8.0 × 10^6^ cells/mL.

**Figure 6 vaccines-09-01006-f006:**
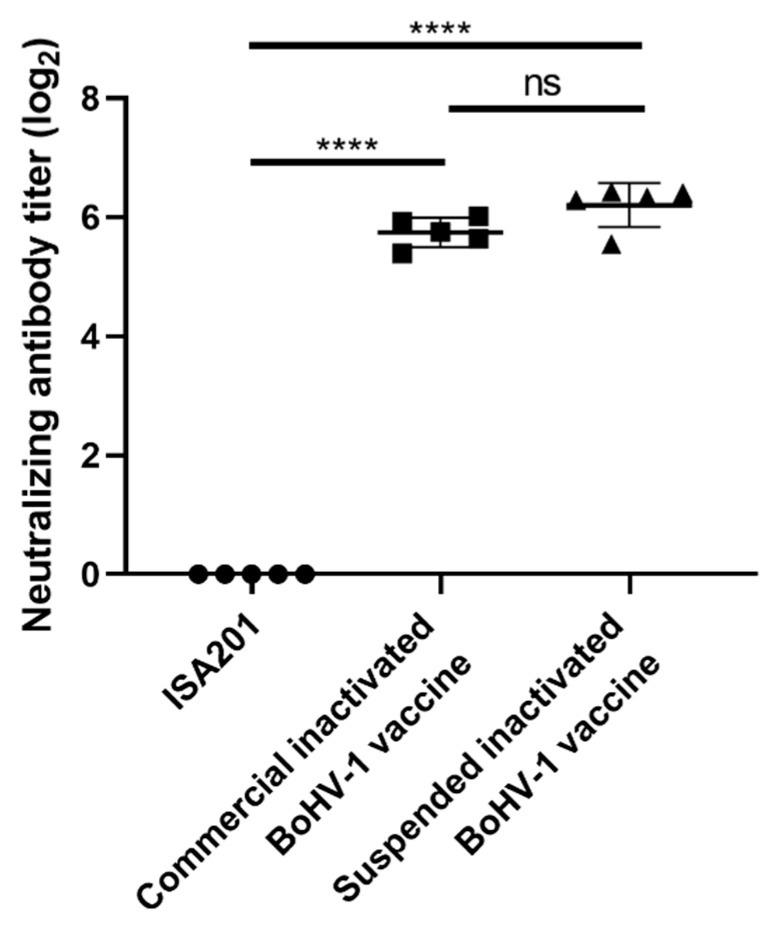
The neutralizing antibodies induced by ISA201, BoHV-1 inactivated vaccine manufactured in suspended MDBK cells and commercial inactivated BoHV-1 vaccine, significance code: “****” representing *p* < 0.0001, “ns” for no significant difference.

## Data Availability

The data presented in this study are available in article.

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
