# Peer review of "Establishment of a Suspension MDBK Cell Line in Serum-Free Medium for Production of Bovine Alphaherpesvirus-1"

_vaccines, 2021, doi:10.3390/vaccines9091006_

Round 1

Reviewer 1 Report

The authors present an interesting study which describes the adaptation of the widely used Madin-Darby bovine kidney (MDBK) cells for suspension growth in serum free media. The authors subsequently investigated the performance of these cells to support the growth of bovine alphaherpesvirus type 1 (BoHV-1).

The immune responses to inactivated BoHV-1 produced under these conditions was then compared to the immune responses of a commercial BoHV-1 vaccine in a guinea pig model.

Overall, the conclusions drawn are well supported by the presented data. I believe the study will be of interest those working in this field. Virus nomenclature: Throughout the manuscript the virus of interest is referred to as infectious bovine rhinotracheitis virus (IBRV) which is one of the historical names for of bovine alphaherpesvirus 1 (BoHV-1). There are several historical names for BoHV-1 which are associated with the diseases the virus has been isolated from, in this manuscript the strain used was most likely isolated from a case of infectious bovine rhinotracheitis (IBR).

However, I believe it is widely accepted now that these viruses are all the same species and as such, I would encourage the authors to adopt the ICTV nomenclature throughout out their manuscript. The potential folly of using the disease associated nomenclature is demonstrated on one of the rationales for the study, namely, the involvement of IBRV in bovine respiratory disease (BRD).

Suggest and comments for the authors to consider:

Line 13 suggest revision “this study was to”

Line 15 suggest revision “and chose ST503 for subsequent experiments.”

Line 22 suggest revision “enhance virus yield 11-fold, with the virus titre reaching”

Line 23 suggest revision “the prototype inactivated BoHV-1 vaccine we generated using the suspension cultures of MDBK cells” As the authors have not demonstrated that their immunisation formulation can protect from disease the use of the term “vaccine” should be qualified.

Line 24 Please review the statement “induced a higher level of neutralizing antibodies than commercial inactivated IBRV vaccine.” This statement is arguably exaggerated compared to the statement made in describing the relevant data shown in Figure 6.

On lines 332 to 334 in the results section, the authors state: “Importantly, inactivated IBRV made from the suspended MDBK cells in lab using the protocol described above elicited neutralization antibodies to a titer comparable to that of the commercial vaccine, suggesting decent immunogenicity.” While the data in Figure 6 suggests, the average antibody titres for the virus grown in suspension cultures was higher than those elicited by the commercial vaccine, the differences were statistically significant. The relevant text in the abstract should be modified to ensure this is clear.

Line 41 suggest revision “Madin-Darby bovine kidney (MDBK)”

Line 51 suggest revision “BoHV-1 is one of the viruses which is strongly associated with bovine” Causation can be very difficult to attribute, particularly in respect to BRD. Line 60 suggest revision “we adapted adherent MDBK cells”

Line 51 suggest revision “Gibco”

Line 148 How was the filtration performed, e.g. what pore size?

Line 177 suggest revision “for their capacity to support of MDBK cell growth”

Line 188 suggest revision “In comparison, the growth of cells”

Line 193 Figure 1 What to the scale bars represent on Fig 1A to 1E? Line 206 suggest revision “began to recover after passage 4”

Line 226 suggest revision “rounded and buoyant”

Line 265 Figure 4 – A scale or magnification should be provided for each panel. Line 268 suggest revision “by microscope.”

Line 298-307 The interpretation of Fig. 5G. I am not sure that I understand this section, partly, as the adherent cell numbers are referred as being infected at a “cell density” – as they are adherent then I presume the “density” refers to the concentration at which they were added to the flask. Is this correct? If so, is this relevant? Add to this as the authors have used an MOI = 0.1 for all three cultures, more virus would have been added to the third culture with the highest number of cells, it therefore stands to reason it would yield more virus. I guess I am wondering if total virus yield would be a more relevant comparator here? Where any statistical comparisons done on these data?

Line 309 Figure 5 For Figure 5A to 5D and 5G– “lg” in the titles of the y-axes should be “Log10” while this is stated in the figure legend, it should also be clear on the actual figures.

Line 352 While the authors have demonstrated the capacity to adapt MDBKs to suspension cultures in a reduced timeframe compared to the cited reference, that study utilised MDCK, so the authors confident that their processes would be directly translatable to a different cell type?

Lines 355 Are all of the studies which reported the adaptation of MDBK cells to serum free media and suspension culture cited here? It is essential that the authors place their study in the context of the existing literature. These few sentences (lines 355 to 362) only contain one direct comparison to that of Hao et al. [27]. Which raises the questions of what was different and what was common between these studies and the current one? In general, the discussion tends to paraphrase the results, rather than placing them into the context of the relevant literature. Please review.

Line 428 suggest revision by replacing “In a word” with “In conclusion”. More than one word follows.

Line 524 – is this reference available in an accessible repository?

Author Response

Dear Reviewer:

              We have studied your valuable comments and suggestions carefully, and responded your comments point by point in the attachment.

Yours sincerely,

Pengpeng Wang

Response to Reviewer 1 Comments

Point 1: Virus nomenclature: Throughout the manuscript the virus of interest is referred to as infectious bovine rhinotracheitis virus (IBRV) which is one of the historical names for of bovine alphaherpesvirus 1 (BoHV-1). There are several historical names for BoHV-1 which are associated with the diseases the virus has been isolated from, in this manuscript the strain used was most likely isolated from a case of infectious bovine rhinotracheitis (IBR).

Response 1: Thank you for your valuable advice. According to your suggestion, we reconsidered the name of the virus. The virus strain used in our experiment was isolated from a clinical case with respiratory diseases. In the scientific research of the virus, the name of bovine alphaherpesvirus 1 is obviously more authoritative. Therefore, we changed the name of the virus in this manuscript to bovine alphaherpesvirus 1 (BoHV-1).

Point2: Moderate English changes required: Line 13 suggest revision “this study was to”; Line 15 suggest revision “and chose ST503 for subsequent experiments.”; Line 22 suggest revision “enhance virus yield 11-fold, with the virus titre reaching”; Line 23 suggest revision “the prototype inactivated BoHV-1 vaccine we generated using the suspension cultures of MDBK cells” As the authors have not demonstrated that their immunisation formulation can protect from disease the use of the term “vaccine” should be qualified; Line 41 suggest revision “Madin-Darby bovine kidney (MDBK)”; Line 51 suggest revision “BoHV-1 is one of the viruses which is strongly associated with bovine” Causation can be very difficult to attribute, particularly in respect to BRD; Line 60 suggest revision “we adapted adherent MDBK cells”; Line 51 suggest revision “Gibco”; Line 177 suggest revision “for their capacity to support of MDBK cell growth”; Line 188 suggest revision “In comparison, the growth of cells”; Line 206 suggest revision “began to recover after passage 4”; Line 226 suggest revision “rounded and buoyant”; Line 428 suggest revision by replacing “In a word” with “In conclusion”. More than one word follows.

Response 2: According to your comments, we have corrected and modified the sentences again. And we found that the reviewer’s suggestions on the grammar and sentence structure in our manuscript made up for our defects in English expression, and we have modified the corresponding parts in the manuscript. Thank you very much for your valuable suggestions.

Point 3: Line 24 Please review the statement “induced a higher level of neutralizing antibodies than commercial inactivated IBRV vaccine.” This statement is arguably exaggerated compared to the statement made in describing the relevant data shown in Figure 6. On lines 332 to 334 in the results section, the authors state: “Importantly, inactivated IBRV made from the suspended MDBK cells in lab using the protocol described above elicited neutralization antibodies to a titer comparable to that of the commercial vaccine, suggesting decent immunogenicity.” While the data in Figure 6 suggests, the average antibody titres for the virus grown in suspension cultures was higher than those elicited by the commercial vaccine, the differences were statistically significant. The relevant text in the abstract should be modified to ensure this is clear.

Response 3: We are sorry for this vague description. In the abstract, we wanted to show that although there was no significant difference between the two groups of data, the BoHV-1 inactivated vaccine manufactured in suspended MDBK cells tended to induce higher levels of neutralizing antibodies than that the commercial vaccine (Fig. 6). Now it seems that this description is not accurate enough. We have modified it to “Most importantly, the prototype inactivated BoHV-1 vaccine we generated using the suspension cultures of MDBK cells induced neutralizing antibodies to a titer comparable to that of the commercial inactivated BoHV-1 vaccine” in the abstract. Thank you very much for your comments.

Point 4: Line 148 how was the filtration performed, e.g. what pore size?

Response 4: Thank you very much for pointing out our problem. It was our negligence that the filtration method was not clearly described. This method was “After repeated freeze-thaw cycles or ultrasonic treatment, the treated virus suspension was centrifuged at 3000 rpm for 5 min and the supernatant was filtered by 0.2 μm filter for virus titer determination”. And we have added this method to our manuscript.

Point 5: Line 193 Figure 1 What to the scale bars represent on Fig 1A to 1E?

Response 5: The scale bar in the background of the Fig. 1A to E is the grid used for cell counting. In the early stage of suspension adaptation process, the cell counting chamber was used to observe cell aggregation and count the number of single cells. In our opinion, although these grids clutter the figure background, it does not affect the observation of cell state.

Point 6: Line 265 Figure 4 – A scale or magnification should be provided for each panel.

Response 6: According to your comments, we have added a scale bar to each panel, which makes our results more scientific and rigorous. Thank you for your valuable comments.

Point 7: Line 298-307 The interpretation of Fig. 5G. I am not sure that I understand this section, partly, as the adherent cell numbers are referred as being infected at a “cell density” – as they are adherent then I presume the “density” refers to the concentration at which they were added to the flask. Is this correct? If so, is this relevant? Add to this as the authors have used an MOI = 0.1 for all three cultures, more virus would have been added to the third culture with the highest number of cells, it therefore stands to reason it would yield more virus. I guess I am wondering if total virus yield would be a more relevant comparator here? Where any statistical comparisons done on these data?

Response 7: Thank you for your thoughtful comments. That’s exactly as you described it. In order to evaluate the volumetric virus productivity of adherent and suspension cells, we referred to the method established by C.F. Shen et al [1]., the adherent cells (0.9× 106 cells/mL) in the last phase of logarithmic growth phase were infected with BoHV-1, and the suspension cells with the same culture volume and density were infected with BoHV-1. Therefore, we believe that the virus yield obtained from different cell lines could reflect the volumetric virus productivity of adherent and suspension cells under the same cell concentration, culture volume and MOI. In addition, the third culture was to prove that the virus yield could improve with the enhancement of suspension cell concentration, and which can make up for the lack of production capacity of suspended cells. Therefore, we referred to the method established by V. Dill et al. [2], the suspended cells with different concentrations were infected virus at the same MOI, and the effect of different cell concentration on virus yield were evaluated by comparing the total virus yield. Due to the addition of more virus seeds, there will be defects in taking the total virus yield as the evaluation standard, but it can still reflect the increasing trend to a certain extent. We will pay attention to this problem in subsequent research and actively look for other evaluation indicators to avoid this impact.

References:
[1] C.F. Shen; C. Guilbault; X. Li; S.M. Elahi; S. Ansorge; A. Kamen; R. Gilbert. Development of suspension adapted Vero cell culture process technology for production of viral vaccines. Vaccine 2019, 37, 6996-7002, doi:10.1016/j. vaccine.2019.07.003.
[2] V. Dill; J. Ehret; A. Zimmer; M. Beer; M. Eschbaumer. Cell Density Effects in Different Cell Culture Media and Their Impact on the Propagation of Foot-And-Mouth Disease Virus. Viruses 2019, 11, 511, doi:10.3390/v11060511.

Point 8: Line 309 Figure 5 For Figure 5A to 5D and 5G– “lg” in the titles of the y-axes should be “Log10” while this is stated in the figure legend, it should also be clear on the actual figures.

Response 8: According to your suggestion, we have modified the “lg” to “log10” in the Fig. 5A to 5D and 5G. Thank you for your careful reading of our manuscript. We have corrected it in the new manuscript.

Point 9: Line 352 While the authors have demonstrated the capacity to adapt MDBKs to suspension cultures in a reduced timeframe compared to the cited reference, that study utilised MDCK, so the authors confident that their processes would be directly translatable to a different cell type?

Response 9: Thank you very much to point out this problem. We thought about it seriously. As you pointed out, it’s not rigorous to use different cell types for comparison. We have deleted this reference and highlighted the advantages of the two-step process in our study by comparing the three-step adaptation method established by M.S. Sinacore et al.. The modified content is as follows: “Compared with the method established by M.S. Sinacore et al., the two-step process in our study can save the time of suspension adaptation, and simplify the process of suspension adaptation, which can retain the production capacity of MDBK cells and save economic costs”. Thank you again.

Point 10: Lines 355 Are all of the studies which reported the adaptation of MDBK cells to serum free media and suspension culture cited here? It is essential that the authors place their study in the context of the existing literature. These few sentences (lines 355 to 362) only contain one direct comparison to that of Hao et al. [27]. Which raises the questions of what was different and what was common between these studies and the current one? In general, the discussion tends to paraphrase the results, rather than placing them into the context of the relevant literature. Please review.

Response 10: Thank you for your instructive suggestions. Currently, we can find only two studies about the adaptation of MDBK cells to serum-free and suspension culture. We have reorganized this part according to your suggestions, and the common and differences between these studies and our study were compared. The modified content is as follows: “Several authors, including S. Cai et al. and P. Hao et al. had used CD medium from Gibco and CD MDBK 249 medium from JS Biosciences for suspension adaptation of MDBK cells, respectively. Therefore, we screened out the ST503 which is more suitable for suspension growth of MDBK cells than the other three media including CD MDBK 249 medium (Figure 1). And the maximum concentration of suspension-adapted MDBK cells obtained from P. Hao et al. and S. Cai et al. could only reach 4.0 × 106 cells/mL and 1.25 × 107 cells/mL, respectively. While the density of suspended MDBK cells could reach to 2.5 × 107 cells/mL after making up for the effects of nutrient deficiency (Figure 3C). These results indicate that the suspended MDBK cells display a robust proliferative potential”. Thank you for your valuable advices.

Point 11: Line 524 – is this reference available in an accessible repository?

Response 11: Thank you for your attention. This reference is a master's thesis, which was included in the Dissertation Library of China Agricultural University. Because our research draws on some of its results, we list it as a reference with a scientific and rigorous attitude.

Reviewer 2 Report

The manuscript entitled “Establishment of A Suspension MDBK cell line in Serum Free Medium for Production of Infectious Bovine Rhinotracheitis Virus” is well structured and conducted with extreme scientific rigor by the Authors in each of its sections.

I find no conceptual errors or inaccuracies of any kind in the experimental design which is flawless.

However, the "introduction" section could be expanded by including a few more quotes inherent to IBRV around the world and not just in China.

Instead, the results and discussion are very detailed and I do not believe they need to be further implemented or expanded upon.

Ultimately, I simply recommend an integration of the introductory section with a few more bibliographical references and I congratulate the Authors for the splendid work performed.

Author Response

Dear Reviewer:

            Thank you very much for your comments and suggestions. We are sorry for not clearly and comprehensively describing the epidemic of the disease. And we have supplemented the positive rates of the disease in different areas and its negative effects to the “Introduction” section. The modified content is as follows: “Bovine respiratory disease complex (BRDC) is a general term for cattle respiratory diseases caused by a variety of viruses, bacteria and other factors [13]. Because BRDC is a multifactorial disease, which is often persistent and difficult to resolve due to the unique immunosuppressive and immune avoidant mechanisms exhibited by each pathogen, and co-infection of multiple pathogens can enhance the toxicity and persistence of viral and bacterial [14]. Bovine alphaherpesvirus 1 (BoHV-1) is one of the viruses which is strongly associated with bovine respiratory disease complex (BRDC), which affects cattle health worldwide and causes enormous economic losses in cattle industry [15]. The virus has been widely spread and reported cases are increasing rapidly since the virus was first isolated in 1956 [16]. Studies have shown that the positive rates of BoHV-1 antibody in cattle herds in Ireland, China and Brazil could reach 75% - 90%, 37.9% and 84% respectively, and milk yield of the cattle infected with BoHV-1 will be reduced by nearly 250.9 L per year [17-20]. The virus can cause a variety of clinical symptoms in cattle, including infectious bovine rhinotracheitis (IBR), conjunctivitis and abortion. Vaccination remains one of the most effective strategies to prevent and control the disease”. Thank you for your instructive suggestions again.

Yours sincerely,

Pengpeng Wang

Reviewer 3 Report

Dear Editor,

With interest I read the paper by Wang et al entitled "Establishment of A Suspension MDBK cell line in Serum Free Medium for Production of Infectious Bovine Rhinotracheitis Virus" I made the following comments that I annotated in the attached pdf file.

One major remark is that authors do not mention anything about Bovine neonatal pancytopenia, a pathology induced in cattle that are immunized with vaccines derived from MDBK cultures. Surely something that the authors should mention and provided some scientific prove if this is also holds true for the newly developed cell line the authors describe.

I would reconsider the paper after the authors made the necessary revisions.

Author Response

Dear Reviewer:

            We are very appreciated for your suggestions and comments, and have replied to your comments. Please see below:

Response to Reviewer 3 Comments

Point 1: One major remark is that authors do not mention anything about Bovine neonatal pancytopenia, a pathology induced in cattle that are immunized with vaccines derived from MDBK cultures. Surely something that the authors should mention and provided some scientific prove if this is also holds true for the newly developed cell line the authors describe.

Response 1: Bovine neonatal pancytopenia (BNP) is a alloimmune syndrome caused by vaccine-induced alloreactive antibodies. There are some studies have shown that the MDBK cell line is presumably the origin of BNP associated alloreactivity. However, not all vaccines produced with MDBK cells can cause BNP, the vaccine leading to BNP may be due to the high efficiency and stable conformation of adjuvant, which improves the immunogenicity of MDBK cells. But in any case, the negative effects of MDBK cell line pose a great challenge for antigen purification and adjuvant screening in the downstream production process of vaccine. The suspended MDBK cell line we established is to be used for large-scale culture of vaccine, and the purification of antigen and screening of adjuvant are important technological steps that need to be seriously considered in the next research. We will take whether BNP can be caused by vaccine as an important index of the vaccine produced by this cell line in the follow-up research. Thank you very much for your thoughtful comments.

References:

  1. M. Bastian; M. Holsteg; H. Hanke-Robinson; K. Duchow; K. Cussler. Bovine Neonatal Pancytopenia: is this alloimmune syndrome caused by vaccine-induced alloreactive antibodies? Vaccine 2011, 29, 5267-5275, doi:10.1016/j.vaccine.2011.05.012.
  2. L. Benedictus; C.R. Bell. The risks of using allogeneic cell lines for vaccine production: the example of Bovine Neonatal Pancytopenia. Expert Rev Vaccines 2017, 16, 65-71, doi:10.1080/14760584.2017.1249859.

Point 2: Line 19 suggest revision “18 ± 1μm”

Response 2: Thank you for your valuable suggestion. We have modified the “18.3 ± 1.3”and “16.5 ± 0.91” to “18 ± 1” and “16 ± 1”.

Point 3: Did the authors sequence the genome of the parent MDBK cell line and the suspension cell line? If yes, is the obtained cell line genetically stable and related to the parent cell line? Where these cell lines subjected to Cell line authentication? See for example: Yu M, Selvaraj SK, Liang-Chu MM, Aghajani S, Busse M, Yuan J, Lee G, Peale F, Klijn C, Bourgon R, Kaminker JS, Neve RM. A resource for cell line authentication, annotation and quality control. Nature. 2015 Apr 16;520(7547):307-11. doi: 10.1038/nature14397. PMID: 25877200. How long were the cell lines obtained from CVCC kept in culture prior to initiating this research project? During that time were the cell line authenticated? Was the suspension cell line authenticated and did this match with the parent cell line?

Response 3: Gene sequencing and authentication of cell lines are particularly important for the establishment of new cell lines and the prevention of cross contamination between cell lines. At present, our manuscript has studied the biological characteristics and production capacity of the suspended MDBK cells. And we will authenticate the suspended MDBK cell line and its parental cell line and analyze their genetic stability in subsequent research, so that the suspended MDBK cell line we established can be applied to the production of vaccines and the other biological products. Thank you very much for your valuable comments.

Point 4: Can the authors add the product numbers of the cell lines in the CVCC collection?

Response 4: Thank you very much for pointing out our problem. We have added relevant information in line 78 of the manuscript.

Point 5: Will the suspension cell line developed in this publication be deposited in the CVCC or other collection?

Response 5: After the suspension cell line has been further authenticated and verified, we certainly hope that it can be deposited by CVCC or other institutions.

Point 6: Will it be made available to all who request this cell line?

Response 6: Of course. We hope that the suspension cell line we established can be applied to more research and fields.

Point 7: Also statistically analyze these two groups (Fig.6)?

Response 7: It was our negligence that the significance analysis was not clearly described and shown. And we have added the significance analysis in the Fig.6, and rewritten this part in our manuscript. Thank you very much again.

Point 8: Line 428 suggest deletion “In a word”.

Response 8: According to your suggestion, we have modified the “In a word” to “In conclusion”. Thank you for your careful reading.

Point 9: In the “Conclusion” section, Please, specify and quantify. Statement is not scientific. Reformulate and try to highlight more the impact that the cell line will have on both research and production process?

Response 9: We are sorry for the vague description in the conclusion section, and have quantified the “growth potential of cells” and “virus production”. Meanwhile, we also put forward new ideas for the application of suspension cell lines. Thank you very much for your instructive suggestions.

Reviewer 4 Report

AUTHORS

Title: Establishment of A Suspension MDBK cell line in Serum Free Medium for Production of Infectious Bovine Rhinotracheitis Virus

This is a short research paper on vaccine manufacturing. Since suspension cells are preferred over adherent ones, authors established a procedure for the culture of Infectious Bovine Rhinotracheitis Virus in suspended Madin-Darby Bovine Kidney cell line in serum-free medium. This manuscript is well written and concise, and authors have to be given credit for ascertaining what appears to be a good option for scaling up this procedure. I honestly have very little negative points to highlight and for all the above I advise minor revisions.

Line 51. Can authors better describe the multifactorial etiology of the bovine respiratory disease complex.

Line 52-53. Please add on the impact (morbidity/mortality data, maybe also economic impact data)

Line 56. “Within” cells or “in” cells

Is the SZH strain of IBRV better characterized elsewhere? Please add description when first mentioned

Please add description of multiplicity of infection (MOI) when first mentioned

In the conclusions, the paper might get higher visibility if mentioning that these could be the first steps in using this cell line suspension culture system in the replication of other viruses.

Author Response

Dear Reviewer:

            We are grateful to you for your comments and suggestions, and have tried our best to revise the manuscript. The responses to your comments are listed in the attachment.

Response to Reviewer 4 Comments

Point 1: Line 51. Can authors better describe the multifactorial etiology of the bovine respiratory disease complex.

Response 1: Thank you for your suggestion. We have described in more detail the multifactorial etiology of the bovine respiratory disease complex, and the modified content is as follows: “Bovine respiratory disease complex (BRDC) is a general term for cattle respiratory diseases caused by a variety of viruses, bacteria and other factors [13]. Owing to BRDC is a multifactorial disease, which is often persistent and difficult to resolve due to the unique immunosuppressive and immune avoidant mechanisms exhibited by each pathogen, and co-infection of multiple pathogens can enhance the toxicity and persistence of viral and bacterial [14]”.

Point 2: Line 52-53. Please add on the impact (morbidity/mortality data, maybe also economic impact data)

Response 2: According to your helpful suggestion, we have supplemented the epidemiological information related to the disease in order to more clearly show its negative impact on the cattle industry. And the modified content is as follows: “Bovine alphaherpesvirus 1 (BoHV-1) is one of the viruses which is strongly associated with bovine respiratory disease complex (BRDC), which affects cattle health worldwide and causes enormous economic losses in cattle industry [15]. The virus has been widely spread and reported cases are increasing rapidly since the virus was first isolated in 1956 [16]. Studies have shown that the positive rates of BoHV-1 antibody in cattle herds in Ireland, China and Brazil could reach 75% - 90%, 37.9% and 84% respectively, and milk yield of the cattle infected with BoHV-1 will be reduced by nearly 250.9 L per year [17-20]. The virus can cause a variety of clinical symptoms in cattle, including infectious bovine rhinotracheitis (IBR), conjunctivitis and abortion. Vaccination remains one of the most effective strategies to prevent and control the disease”. Thank you for your valuable advices.

Point 3: Line 56. “Within” cells or “in” cells

Response 3: Thank you very much to point out this problem. We have careful rechecked and corrected the sentences and grammar.

Point 4: Is the SZH strain of IBRV better characterized elsewhere? Please add description when first mentioned

Response 4: Thank you for your valuable suggestion. Based on the previous study, we have added a description of the characteristics of the virus strain in line 145 - 147 of the manuscript.

Point 5: Please add description of multiplicity of infection (MOI) when first mentioned

Response 5: We are sorry for our negligence. We have added the description of multiplicity of infection (MOI) in line 154 of the manuscript.

Point 6: In the conclusions, the paper might get higher visibility if mentioning that these could be the first steps in using this cell line suspension culture system in the replication of other viruses.

Response 6: We really agree with your viewpoint, we have rewritten the conclusion section and look forward to providing insights for the relevant research in this field. Thank you for your instructive suggestions again.

This manuscript is a resubmission of an earlier submission. The following is a list of the peer review reports and author responses from that submission.